# Inhibiting *UNC13B* Suppresses Cell Proliferation by Upregulating the Apoptotic Pathway in Multiple Myeloma

**DOI:** 10.3390/biomedicines13092086

**Published:** 2025-08-27

**Authors:** Yuan Tao, Lihua Yuan, Yuntian Ding, Rongli Xie, Fangjie Liu, Zhongming Zhang, Xiaojun Xu, Xiaobo Wang

**Affiliations:** 1Department of Hematology, The Affiliated Panyu Central Hospital of Guangzhou Medical University, 8 Fuyu East Road, Panyu, Guangzhou 511400, China; taoyuan@pyhospital.com.cn; 2Division of Pediatric Surgery, Department of Surgery, The University of Hong Kong Shenzhen Hospital, Shenzhen 518106, China; yuanlh@hku-szh.org; 3Department of Hematology, The Seventh Affiliated Hospital of Sun Yat-sen University, Shenzhen 518106, China; dingyuntian@sysush.com (Y.D.); jierongli@sysush.com (R.X.); liufangjie@sysush.com (F.L.); 4Department of Hematology, The First Affiliated Hospital of Guangxi Medical University, Nanning 530021, China; zzmmissyou@126.com

**Keywords:** multiple myeloma, UNC13B, cell proliferation, apoptosis, cell cycle regulation, PINK1, CDK2, AKR7A3, RNA interference, bioinformatics analysis

## Abstract

**Background/Objectives:** Multiple myeloma (MM) is the second most common hematological malignancy and remains incurable because of its complex and heterogeneous pathogenesis. *UNC13B* (unc-13 homolog B) encodes Munc13-2, a presynaptic protein that is involved in vesicle exocytosis. While its role has been explored in neurological diseases, its function in cancer biology remains largely uncharacterized. This study aimed to elucidate the role of UNC13B in regulating MM cell proliferation and apoptosis. **Methods:**
*UNC13B* mRNA expression was assessed across human MM cell lines. ARD cells, which exhibited the highest *UNC13B* expression, were transduced with a UNC13B-specific shRNA via a lentiviral vector. Cell proliferation, apoptosis, and expression of associated proteins were evaluated by means of the Cell Counting Kit-8 (CCK-8) assay, flow cytometry, and Western blot analysis. **Results:** *UNC13B* was significantly upregulated in MM cell lines. The knockdown of *UNC13B* in ARD cells markedly inhibited cell proliferation and induced apoptosis. These changes were accompanied by the downregulation of proliferation-related proteins and upregulation of pro-apoptotic markers. Western blot analysis suggests that UNC13B may exert its effects by modulating key regulatory proteins, including PINK1, CDK2, AKR7A3, and Bim. **Conclusions:** Our findings suggest that UNC13B supports MM cell survival and proliferation, potentially through the regulation of oncogenic and apoptotic signaling pathways. UNC13B may represent a novel therapeutic target in multiple myeloma.

## 1. Introduction

Multiple myeloma (MM) is a malignant clonal disorder of plasma cells characterized by the secretion of monoclonal immunoglobulins, leading to end-organ damage such as anemia, renal dysfunction, osteolytic bone lesions, and hypercalcemia [1,2]. Between 1990 and 2016, the global incidence of MM increased by approximately 126%, making it the second most prevalent hematologic malignancy after non-Hodgkin lymphoma and accounting for nearly 10% of all hematologic cancers [3]. The median age at diagnosis is approximately 69 years, with over 60% of cases occurring in individuals aged 65 and older [4]. Although the introduction of novel therapeutic agents and the widespread application of autologous stem cell transplantation have extended the median overall survival to 7–8 years [5], MM remains incurable, and the majority of patients eventually experience disease relapse [6]. These challenges underscore the urgent need for novel therapeutic strategies targeting the molecular drivers of MM pathogenesis.

Although the precise etiology of MM remains incompletely defined, genomic instability is recognized as a hallmark feature of the disease [7]. MM typically evolves from asymptomatic precursor stages, beginning with monoclonal gammopathy of undetermined significance (MGUS), progressing to smoldering multiple myeloma (SMM), and ultimately developing into symptomatic MM [1,2]. MGUS is considered the earliest premalignant phase, with an annual progression risk of approximately 1% [8]. SMM represents an intermediate asymptomatic stage, characterized by a greater clonal plasma cell burden in the bone marrow but without clinical end-organ damage, and carries an annual progression risk of approximately 10% [9]. This stepwise transformation underscores the continuum of malignant evolution in MM. Multiple biological processes contribute to this progression, including aberrant gene expression, structural chromosomal abnormalities [10,11], dysregulated signaling pathways, epigenetic modifications [12], alterations in the bone marrow microenvironment [13], and immune system dysfunction [14,15].

*UNC13B*, located on chromosome 9p13.3, spans approximately 243 kilobases and consists of 39 exons encoding the presynaptic protein Munc13-2, which comprises 1591 amino acids [16]. Munc13-2 is primarily known for its role in regulating early steps of exocytosis at presynaptic terminals by modulating calcium channel activity [17]. Accordingly, research on UNC13B has largely focused on the fields of neurology and psychiatry, where mutations have been implicated in disorders such as bipolar disorder [18] and epilepsy [19,20]. Exocytosis, the process by which cells secrete bioactive molecules, is essential not only for normal physiological processes but also for pathological conditions [21]. In cancer, tumor cells exploit exocytosis to release factors into the extracellular environment, modulating the tumor microenvironment to promote tumorigenesis, invasion, and metastasis [21,22,23]. Accumulating evidence suggests that exocytosis is a critical mechanism underlying aggressive cancer behavior, and its regulatory components may serve as promising therapeutic targets [24].

In our previous study, gene expression profiling and functional validation demonstrated that *UNC13B* was significantly upregulated in arsenic trioxide (ATO)-resistant K-562 chronic myeloid leukemia (CML) cells and that it promoted resistance by enhancing cell survival and suppressing apoptosis [25]. These findings suggest that UNC13B plays a role in the regulation of cell fate under therapeutic stress. However, its function in other hematologic malignancies remains unclear. Given the critical importance of apoptotic dysregulation and treatment resistance in multiple myeloma (MM), we hypothesized that UNC13B may also contribute to MM pathogenesis. Therefore, this study aimed to investigate the expression and biological role of UNC13B in MM cells and to elucidate the potential mechanisms by which it regulates cell proliferation and apoptosis. By addressing this question, our work aims to fill a critical gap in the understanding of exocytosis-related proteins in MM pathogenesis and identify potential molecular targets for therapeutic intervention.

## 2. Materials and Methods

### 2.1. Cell Culture

Human multiple myeloma cell lines ARD (CL-0778), U266 (CL-0510), and RPMI 8226 (CL-0564) were obtained from Procell Life Science & Technology Co., Ltd. (Wuhan, China). All cell lines were cultured in RPMI-1640 medium (Gibco, Thermo Fisher Scientific, Shanghai, China; Cat#11875-093) supplemented with 10% fetal bovine serum (FBS, Gibco, Cat#10099-141) and 1% penicillin–streptomycin solution (Gibco, Cat#15140-122) at 37 °C in a humidified incubator containing 5% CO_2_.

### 2.2. Cell Transfection and Selection

To investigate the functional role of UNC13B, two short hairpin RNA (shRNA) sequences targeting *UNC13B* (GenBank accession no.: NM_020313.2) were designed and cloned into the lentiviral vector lentiCRISPR v2 (Addgene plasmid #52961). The shRNA target sequences were as follows:

shUNC13B-1: 5′-GACCTACAAATGGGAGCAAAT-3′

shUNC13B-2: 5′-CGAGTCCTATGAGTTGCAGAT-3′

Lentiviral particles were generated by co-transfecting lentiCRISPR v2-shRNA constructs with the packaging plasmid psPAX2 (Addgene #12260) and the envelope plasmid pMD2.G (Addgene #12259) into human embryonic kidney 293T (HEK293T) cells using Lipofectamine 3000 (Thermo Fisher Scientific, Shanghai, China). Viral supernatants were collected at 48 and 72 h post-transfection, filtered (0.45 μm filter, Millipore, Shanghai, China; Cat#SLHV033RS), and used to transduce ARD multiple myeloma cells.

Transduced cells were selected with puromycin (Sigma-Aldrich, Merck, Shanghai, China; Cat#P8833, 2 μg/mL) for 72 h to eliminate non-infected cells, generating a polyclonal pool of puromycin-resistant cells. Cells transduced with a non-targeting scrambled shRNA served as the negative control.

Knockdown efficiency was first assessed at the mRNA level by means of quantitative real-time PCR (qPCR) 72 h post-selection, a time point at which shRNA-mediated suppression is expected to reach near-maximal effect. Subsequent protein expression and functional assays were performed using cells collected at the same time point.

### 2.3. Quantitative Real-Time PCR (qPCR) for UNC13B Expression

qPCR was used to evaluate *UNC13B* mRNA expression in three multiple myeloma cell lines: U266, ARD, and RPMI 8226. Total RNA was extracted using TRIzol reagent (Thermo Fisher, Shanghai, China; Cat#15596026CN), and RNA quality was assessed for purity and integrity (NanoDrop 2000, Thermo Scientific, Shanghai, China). cDNA was synthesized using theEasyScript First-Strand cDNA Synthesis SuperMix kit (TransGen Biotech, Beijing, China; Cat#AE301). Reactions were performed in RNase-free PCR tubes (20 μL per reaction) under the following conditions: 25 °C for 10 min, 42 °C for 30 min, and 85 °C for 5 s.

The primer sequences were as follows:

*UNC13B*-F: 5′-CCAGCTACACAACTCACTGAGG-3′

*UNC13B*-R: 5′-CTGGTCAGCAAATCCACTGTGG-3′

18SN5-F: 5′-ACCCGTTGAACCCCATTCGTGA-3′

18SN5-R: 5′-GCCTCACTAAACCATCCAATCGG-3′

qPCR reactions were performed using SYBR Green Master Mix (Takara, Beijing, China; Cat#RR820A) in RNase-free PCR tubes (20 μL per reaction) under the following conditions: 95 °C for 5 min; then 40 cycles of 95 °C for 15 s and 60 °C for 34 s; followed by melting curve analysis from 60 °C to 95 °C. Relative expression levels were calculated using the 2^-ΔΔCt^ method [26], with 18SN5 as the reference gene.

### 2.4. Cell Proliferation Assay (CCK-8)

Cells in the logarithmic growth phase were trypsinized, resuspended in complete medium, and seeded at a density of 2000 cells/well in 96-well plates (100 μL/well, three replicates per group). After overnight incubation at 37 °C with 5% CO_2_, 10 μL of CCK-8 reagent (Solarbio, Beijing, China; Cat#CA1210) was added to each well 4 h before the end of culture. Optical density at 450 nm (OD450) values were measured using a microplate reader (iMARK; Bio-Rad, Shanghai, China) after shaking the plate for 2–5 min. The assay was performed daily for 5 consecutive days.

### 2.5. Soft Agar Colony Formation Assay

Cells were digested and resuspended in a 0.6% Noble agarose (Sigma, Cat#A5431) base layer prepared with 10% FBS-containing culture medium. A top layer of 0.3% agarose (Invitrogen, Shanghai, China; Cat#16500-500) in 20% FBS-containing medium was overlaid on the base layer. Plates were incubated for 15 days under standard conditions, with regular hydration of the top layer to prevent drying. Colonies were photographed in randomly selected fields upon formation.

### 2.6. Cell Cycle Analysis

Cells were harvested, washed with pre-chilled Hanks’ balanced salt solution (HBSS; D-Hanks) (pH 7.2–7.4), and fixed in 75% ethanol at 4 °C for at least 1 h. After removing the fixative, cells were washed again with D-Hanks and resuspended in staining buffer. RNase A (100 µg/mL; Solarbio, Cat#9001-99-4) was added and incubated at 37 °C for 15 min, followed by staining with propidium iodide (PI, 50 µg/mL; Sigma, Cat#P4170) at 4 °C in the dark for 15 min. Cell cycle distribution was analyzed using CytoFLEX LX (Beckman Coulter, Shanghai, China), and data were processed with FlowJo v10.8.1 (BD).

### 2.7. Apoptosis Assay

Cells were harvested and washed with pre-chilled D-Hanks, then resuspended in 1× binding buffer. Annexin V-APC (10 μL; Procell, Wuhan, China; Cat#P-CA-207) was added to 200 μL of cell suspension and incubated in the dark at room temperature for 15 min. Samples were analyzed using CytoFLEX LX, and apoptosis rates were quantified with FlowJo v10.8.1.

### 2.8. Western Blotting

Proteins were extracted by ultrasonic lysis and quantified using a BCA protein assay kit (Abcam, Cambridge, UK; ab102536). Equal amounts of protein were separated by SDS-PAGE and transferred to polyvinylidene difluoride (PVDF) membranes. Membranes were blocked with 5% bovine serum albumin (BSA) (Vetec, V900933) for 2 h at room temperature and incubated overnight at 4 °C with primary antibodies. After washing, membranes were incubated with HRP-conjugated goat anti-mouse (#7076, 1:10,000) or anti-rabbit (#7074, 1:10,000) secondary antibodies (Cell Signaling Technology, Danvers, MA, USA). GAPDH (#3683, 1:5000) was used as a loading control. Protein bands were visualized using the iBright FL1000 Imaging System (Thermo Fisher), and quantification was performed with iBright Analysis Software (v5.1.0). Primary antibodies included UNC13B (Santa Cruz, sc-136182, 1:1000), cleaved-PARP (CST, #9541, 1:1000), Bax (~21 kDa; CST, #2772, 1:1000), p21 (CST, #2947, 1:1000), PKC-PAN (#9371, 1:1000), PINK1 (#6946, 1:1000), CDK2 (#2546, 1:1000), AKR7A3 (Abcam, ab227231, 1:1000), and Bim (#2933, 1:1000).

### 2.9. Data Analysis

All statistical analyses were performed using GraphPad Prism 9.0 (GraphPad Software Inc., San Diego, CA, USA). Data are expressed as the mean ± standard deviation (SD) from at least three independent experiments. Comparisons between two groups were analyzed using unpaired Student’s *t*-test. One-way ANOVA was used for comparisons among multiple groups. A *p*-value < 0.05 was considered statistically significant.

## 3. Results

### 3.1. Upregulation of UNC13B Gene Expression in Multiple Myeloma Cell Lines

Quantitative real-time PCR (qPCR) was performed to evaluate *UNC13B* mRNA expression in several human multiple myeloma cell lines, including U266, ARD, and RPMI 8226. The results demonstrated that *UNC13B* was upregulated in all three cell lines. Notably, ARD cells exhibited the highest expression level, with a relative *UNC13B* mRNA level of 3.07 ± 0.44 (Figure 1A). Based on this finding, ARD cells were selected for subsequent functional assays to maximize the likelihood of detecting phenotypic changes upon *UNC13B* knockdown.

### 3.2. Downregulation of UNC13B Inhibits Cell Proliferation and Induces Apoptosis in ARD Cells

Following transduction with an UNC13B-targeting shRNA-expressing lentiviral vector, *UNC13B* mRNA expression was significantly reduced in ARD cells compared with the negative control group, confirming successful gene knockdown (Figure 1B). Cell proliferation, assessed by the CCK-8 assay, was markedly decreased in the shUNC13B group relative to controls (*p* < 0.05), indicating that suppression of *UNC13B* impairs ARD cell viability (Figure 2). *UNC13B* knockdown significantly impaired the clonogenic capacity of ARD cells in soft agar. As shown in the representative images (Figure 3A), shUNC13B-transduced cells formed significantly fewer colonies compared with the scrambled control group. Quantitative analysis further confirmed a marked reduction in colony numbers in the shUNC13B group (Figure 3B), indicating that UNC13B contributes to the anchorage-independent growth ability of multiple myeloma cells.

Cell-cycle analysis using PI staining and flow cytometry in ARD cells revealed that *UNC13B* knockdown altered cell-cycle distribution: the proportion of cells in the G1 phase was significantly reduced (*p* < 0.0001), while the proportion of cells in the S phase was significantly increased (*p* < 0.0001), and no statistically significant change was observed in the G2/M phase. These results suggest that *UNC13B* knockdown induces S-phase arrest (Figure 4). Furthermore, Annexin V-APC staining demonstrated a significant increase in apoptotic ARD cells in the shUNC13B group compared with the control group (*p* < 0.01) (Figure 5A).

### 3.3. UNC13B Modulates the Expression of PINK1, CDK2, AKR7A3, and Bim

Consistent with the observed increase in apoptosis (Figure 5A), Western blot analysis demonstrated that *UNC13B* knockdown led to marked alterations in apoptosis- and signaling-related proteins in ARD cells (Figure 5B). As expected, UNC13B protein levels were reduced in the shUNC13B group compared with the scrambled control, confirming efficient knockdown. Cleaved PARP, a hallmark of caspase-dependent apoptosis, was dramatically elevated, and the pro-apoptotic protein Bax was upregulated. Analysis of additional signaling molecules revealed that the expression of PTEN-induced kinase 1 (PINK1), a serine/threonine protein kinase involved in mitochondrial quality control and stress signaling [27,28,29,30], was decreased following *UNC13B* silencing, suggesting a potential role for UNC13B in maintaining mitochondrial homeostasis via PINK1-dependent pathways. Similarly, the expression of aldo-keto reductase family 7 member A3 (AKR7A3), an enzyme that detoxifies aldehydes and ketones and has been implicated in cancer-related metabolic and signaling pathways [31,32,33,34,35], was also reduced, indicating that AKR7A3 may contribute to the tumor-promoting effects of UNC13B in multiple myeloma. CDK2 expression was decreased, while the pro-apoptotic protein Bim was upregulated, and PKC levels showed only minimal changes. To further assess cell-cycle regulation downstream of UNC13B, we examined p21 (CDKN1A), a canonical inhibitor of the CDK2–Cyclin E/A complex. UNC13B knockdown increased p21 protein levels, consistent with the observed reduction in proliferation and S-phase accumulation.

Collectively, these results indicate that downregulation of *UNC13B* suppresses cell proliferation, disrupts normal cell-cycle progression, and promotes apoptosis in ARD multiple myeloma cells. These effects suggest that UNC13B may play a critical role in maintaining the proliferative and survival capacities of MM cells, likely through modulation of apoptosis-related pathways.

## 4. Discussion

In the nervous system, UNC13B (also known as Munc13-2) is a key regulator of vesicle priming and calcium-triggered exocytosis, determining whether synaptic vesicles can enter a fusion-ready state and complete neurotransmitter release [17,36]. This mechanism is also conserved in large dense-core vesicle (LDCV) exocytosis in non-neuronal secretory cells, where Munc13-2 is essential for vesicle fusion in chromaffin and endocrine cells [37]. These findings suggest that UNC13B broadly controls the efficiency and threshold of vesicle–membrane fusion. Recent studies have extended this concept to cancer biology, where dysregulated vesicular trafficking is increasingly recognized as a driver of tumor progression and therapeutic resistance [21].

In tumor biology, there is substantial evidence that cancer cells exploit vesicular trafficking and secretion to remodel the microenvironment: by releasing soluble cytokines (e.g., IL-6, IL-8, VEGF), growth factors (e.g., TGF-β, EGF), proteases (e.g., MMP2, MMP9), and extracellular vesicles (EVs) carrying integrins, tetraspanins (CD9, CD63, CD81, CD82), and various noncoding RNAs, tumor cells can promote proliferation, invasion, angiogenesis, immune evasion, and therapy resistance [38]. Mechanistically, UNC13B could influence both the total output and the cargo composition of secretory vesicles, thereby shaping the tumor niche and therapeutic response [17,36,37,38]. Our functional data are consistent with this hypothesis: in hematologic tumor models, silencing UNC13B reduced proliferation and enhanced apoptosis, accompanied by the downregulation of mitochondrial quality control (e.g., PINK1) and cell cycle factors (e.g., CDK2), suggesting that UNC13B supports tumor cell survival through a “secretion/exocytosis–mitochondrial stress–cell cycle” axis. Notably, the proteins we selected for detailed analysis—PINK1, CDK2, AKR7A3, and Bim—were chosen to represent four major biological processes potentially regulated by UNC13B: mitochondrial homeostasis, cell cycle control, metabolic adaptation, and intrinsic apoptosis. Among these, mitochondrial quality control is increasingly recognized as a vulnerability in hematologic malignancies [39], suggesting that UNC13B may influence MM cell survival partly through mitochondrial dynamics and mitophagy regulation.

With respect to expression patterns, publicly available datasets do not support a universal “UNC13B overexpression” across cancers; instead, UNC13B is naturally highly expressed in normal brain tissue, consistent with its neuronal function, which can result in apparently higher expression in normal tissue compared to certain tumor types (Tissue expression of UNC13B. Available from https://www.proteinatlas.org/ENSG00000134222-UNC13B/tissue (accessed on 21 August 2025)). This underscores the fact that the oncological relevance of UNC13B likely stems from functional dependency and pathway coupling rather than absolute mRNA abundance. In line with this, the functional perturbation of UNC13B (e.g., knockdown) can elicit marked anti-proliferative and pro-apoptotic effects even in settings without extreme overexpression [40]. From a translational perspective, future work should focus on the role of UNC13B as a node linking secretion and organelle homeostasis, and on its regulation of specific secretory cargos, which may explain the apparent “expression–function” decoupling in different tissue contexts and offer opportunities for precision targeting [37,38].

CDK2 is a cyclin-dependent kinase encoded by the CDK2 gene and is essential for cell cycle progression, particularly DNA replication, damage response, and apoptotic regulation [41]. CDK2 remains active from the late G1 to the S phase and forms complexes with cyclins E and A to drive cells through the G1/S transition [42,43]. Overexpression of CCNE1, the gene encoding cyclin E, is common in various tumor types and contributes to aberrant CDK2 activation, rendering tumor cells dependent on this complex for proliferation [44]. Targeting CDK2 and its cyclin partners has emerged as a promising strategy in cancer therapy, and several CDK2 inhibitors have progressed to clinical trials [45]. In our study, downregulation of UNC13B was associated with decreased CDK2 protein expression. Such vesicular trafficking-mediated regulation of nuclear signaling and cell cycle progression remains an emerging concept, and further mechanistic studies are needed to clarify whether these effects involve autocrine growth factor secretion.

Bim, a member of the Bcl-2 protein family, functions as a pro-apoptotic BH3-only protein. It exerts its pro-apoptotic activity by binding to anti-apoptotic Bcl-2 family members with high affinity, thereby promoting mitochondrial outer membrane permeabilization and caspase activation [46,47]. Bim also plays a role in autophagy by modulating Beclin-1, a key autophagy-related protein [48]. Accumulating evidence suggests that upregulation of Bim enhances tumor cell apoptosis and sensitizes cells to various therapeutic agents [46,49]. In the present study, UNC13B knockdown resulted in elevated Bim expression, indicating that suppression of UNC13B may promote apoptosis in MM cells through the upregulation of Bim.

Protein kinase C (PKC) is a multifunctional serine/threonine kinase involved in diverse cellular processes, including transcriptional regulation, signal transduction, and cell survival [50,51,52], but its role in cancer is context-dependent [53,54,55]. In our study, UNC13B knockdown did not significantly alter PKC levels in ARD cells, suggesting that PKC is not a key downstream effector in this setting.

Collectively, these findings suggest that UNC13B may regulate multiple oncogenic and apoptotic signaling pathways—such as those involving PINK1, CDK2, AKR7A3, and Bim—to promote cell survival and proliferation in multiple myeloma. Therefore, UNC13B represents a promising potential therapeutic target for the treatment of MM.

Importantly, to the best of our knowledge, there are currently no small-molecule inhibitors or therapeutic antibodies specifically targeting UNC13B in clinical use. Nevertheless, emerging preclinical efforts have identified compounds that inhibit related Munc13 family members: for instance, a class of 2-aminobenzothiazoles—termed “bexins”—were shown to selectively block the late stages of Munc13-4-mediated secretory granule fusion in mast cells by interfering with the C2-domain-dependent membrane interactions [56]. While UNC13B possesses distinct domains and tissue distribution, this example demonstrates that targeted inhibition of Munc13-mediated exocytosis is pharmacologically feasible. In the absence of direct inhibitors, indirect targeting strategies might be pursued, such as modulating upstream regulators (e.g., RIM–UNC13 interactions), disrupting functional domains (e.g., the C1/C2 modules), or applying modulators of vesicle priming like phorbol esters and their analogs. Indeed, Munc13-1 has been characterized as a molecular target of bryostatin 1, a C1-domain–binding compound, although its relevance to UNC13B remains to be explored [57]. Our current findings implicate UNC13B as a functionally important node in multiple myeloma, and thus provide a conceptual foundation for future therapeutic development—both in terms of designing UNC13B-specific inhibitors and exploring synergistic strategies that disrupt exocytosis and cell survival.

## 5. Conclusions

Our study demonstrates that *UNC13B* mRNA is upregulated in multiple myeloma cell lines. Knockdown of *UNC13B* significantly suppresses cell proliferation and induces apoptosis in MM cells. Mechanistically, UNC13B may exert tumor-promoting effects by modulating key regulatory proteins, including PINK1, CDK2, AKR7A3, and Bim. Collectively, these findings suggest that UNC13B contributes to MM pathogenesis and may serve as a promising therapeutic target in multiple myeloma.

## Figures and Tables

**Figure 1 biomedicines-13-02086-f001:**
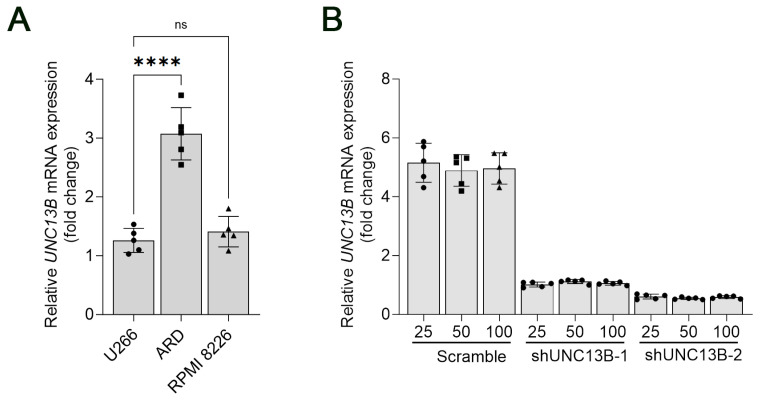
Relative mRNA expression of *UNC13B* in multiple myeloma cell lines. (**A**) Quantitative real-time PCR (qPCR) analysis of *UNC13B* mRNA levels in U266, ARD, and RPMI 8226 cell lines. U266 was used as the reference control for relative quantification, and GAPDH served as the internal reference gene. Expression levels were calculated using the 2^-ΔΔCt^ method and are presented as fold-change relative to U266. The mean ± standard deviation of *UNC13B* expression was 1.00 ± 0.26 in U266, 3.07 ± 0.44 in ARD, and 1.41 ± 0.26 in RPMI 8226 cells. Overlaid symbols (●, ■, ▲) denote individual independent experiments (*n* = 5 per group); different marker shapes are used only for visual distinction and carry no additional meaning. (**B**) Relative *UNC13B* mRNA expression in ARD cells following different treatments. The Scramble + 25 μM treatment group was used as the reference for comparison. **** indicate *p* < 0.0001; ns = not significant (*p* ≥ 0.05). Data are presented as the mean ± standard deviation (SD, *n* = 5 independent experiments).

**Figure 2 biomedicines-13-02086-f002:**
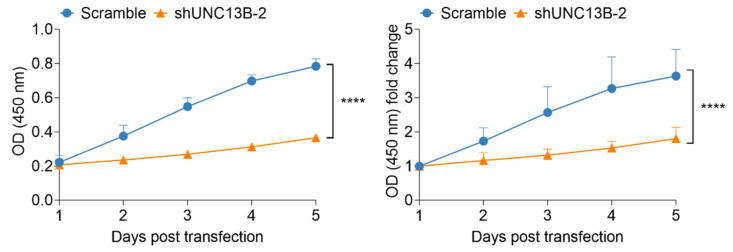
Effect of *UNC13B* gene ablation on cell proliferation. The proliferation rate of ARD cells in the shUNC13B group was significantly inhibited (*p* < 0.0001), suggesting that *UNC13B* expression correlated with the proliferative capacity of ARD cells. **** indicate *p* < 0.0001; Data are presented as mean ± SD (*n* = 3 independent experiments).

**Figure 3 biomedicines-13-02086-f003:**
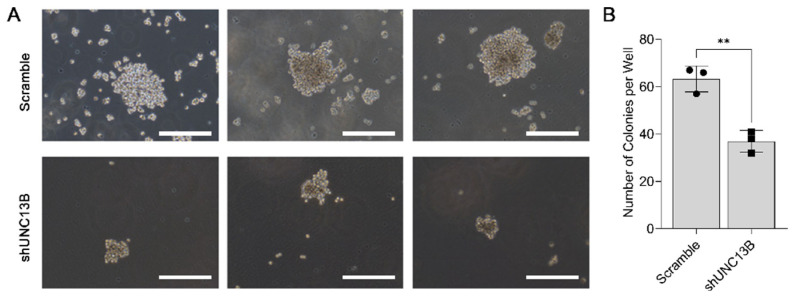
Effect of *UNC13B* gene ablation on the clonogenic ability of cells. (**A**) Representative images of colony formation in soft agar. ARD cells stably transduced with either shUNC13B or scrambled shRNA were seeded into 6-well plates (1000 cells per well) in soft agar consisting of a 0.35% agar upper layer over a 0.6% agar base layer. After 14 days of incubation at 37 °C in 5% CO_2_, visible colonies (>50 µm in diameter) were observed under a microscope. (**B**) Quantification of colony numbers. The total number of colonies per well was recorded.Overlaid symbols (●, ■) denote individual independent experiments; different marker shapes are used only for visual distinction and have no additional meaning. Data are presented as the mean ± SD. Statistical analysis was performed using a two-tailed unpaired *t*-test. ** indicates *p* < 0.01. Scale bar: 500 µm.

**Figure 4 biomedicines-13-02086-f004:**
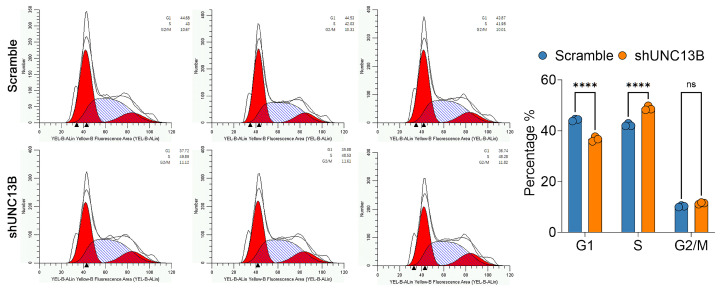
Effect of UNC13B gene ablation on the cell cycle in ARD cells. Five days after transduction with shRNA lentiviral vectors, the shUNC13B group showed a reduced fraction of ARD cells in G1 (*p* < 0.0001) and an increased fraction in S phase (*p* < 0.0001) compared with the Scramble control, with no significant change in G2/M (ns, not significant). Histograms show DNA-content modeling: red areas denote fitted G1 and G2/M peaks, the blue hatched area denotes S phase, and the black line is the measured distribution; ▲ indicates software-generated peak positions (reference only, not used for quantification). **** indicate *p* < 0.0001.

**Figure 5 biomedicines-13-02086-f005:**
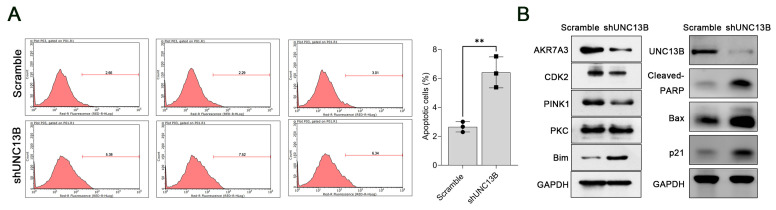
Effect of *UNC13B* gene ablation on apoptosis in ARD cells. (**A**) Flow cytometric analysis of apoptosis using Annexin V-APC staining. The shUNC13B group exhibited a significantly higher percentage of apoptotic ARD cells compared with the scrambled control group (*p* < 0.01); Overlaid symbols (●, ■) denote individual independent experiments; different marker shapes are used only for visual distinction and have no additional meaning. (**B**) Western blot analysis of apoptosis- and signaling-related proteins in ARD cells stably transduced with scrambled shRNA or UNC13B shRNA (shUNC13B) for 72 h. Expression levels of PINK1, CDK2, and AKR7A3 were downregulated, while Bim was upregulated, and PKC levels changed only marginally. In the same samples, UNC13B protein expression was markedly reduced, accompanied by substantial increases in cleaved-PARP (~89 kDa), Bax (~21 kDa), and p21 (CDKN1A) levels. GAPDH served as the loading control. ** indicate *p* < 0.01.

## Data Availability

The data used to support the findings of this study are available from the corresponding author upon request.

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
