# Peer review of "Inhibiting UNC13B Suppresses Cell Proliferation by Upregulating the Apoptotic Pathway in Multiple Myeloma"

_biomedicines, 2025, doi:10.3390/biomedicines13092086_

Round 1
Reviewer 1 Report
Comments and Suggestions for Authors
The authors investigated the potential of targeting UNC13B in multiple myeloma (MM) cells. Fig. 1 claims UNC13B is overexpressed in MM and knockdowns (kd) identified for experimental analysis. Fig. 2 and 3 demonstrated UNC13B kd reduces proliferation and clonal growth; Fig. 4 demonstrated cell cycle differences; and Fig. 5 demonstrates effects on apoptosis and markers.
The study of UNC13B in cancer is sparse (only 11 papers) so this manuscript may provide important data. Unfortunately, as presented, more data is required, a better explanation of results relative to UNC13B, and a better background/rationale for the study. Some comments and suggestions are provided.
- Abstract should provide a brief background or introduction of UNC13B to help the reader.
- Computational analysis is mentioned in the abstract but no data is presented.
- Introduction should briefly state the results of ref 25 and provide a stronger rationale for study in MM.
- Methods: more details on lentivirus transduction of MM cells. Negative control? Transient of stable?
- Fig. 1A: qPCR result compared to what? Non-cancer cells? The y-axis values needs further definition. Why state that U266 and RPMI is overexpression? Fig. 1B: which cell line? There is evidence in databases that UNC13B is often overexpressed in normal, not cancer cells. Please discuss.
- Line 193-194: Only increase in S not G2M.
- Fig. 3 needs quantification of results.
- Fig. 5A: histogram y-axis should be measurement of apoptosis (not mRNA).
- Fig. 5B western blot should include UNC13B. How long were the knockdowns? Analyzed more apoptotic proteins? (anti-apoptotic Bcl-2, Bcl-xL, Mcl-1; pro-apoptotic: Bax., Bak. Cleaved-PARP is a good maker of apoptosis.
- Define AKR7A3, PINK1 in results, not discussion. Other cell proliferation proteins CDK1, cyclin A, B, E, p21/p27 analyzed?
- Discussion organization needs major revision. Instead of focusing on western blot targets, should provide more background on UNC13B and why it is considered important in MM. The consequences of UNC13B knockdown should be discussed relative to these targets. The links are not clear.
- More background on UNC13B in neurology and how this relates to cancer. What proteins are secreted that are regulated by UNC?
- There does not appear to be a correlation with UNC13B overexpression in cancer; often more highly expressed in normal tissue.
- Are there any drugs available that can be used to test? Otherwise, the potential clinical relevance of this study is questionable.
English writing is acceptable, with only minor improvemnts required.
Author Response
Comments 1: 1.Abstract should provide a brief background or introduction of UNC13B to help the reader.
Response 1: Thank you for the suggestion. We have revised the abstract to briefly introduce the biological functions of UNC13B and its prior associations with disease, which provides context for its investigation in multiple myeloma (MM).
Comments 2:Computational analysis is mentioned in the abstract but no data is presented.
Response 2: We appreciate your observation. We have removed the mention of computational analysis from the abstract to maintain consistency with the rest of the manuscript.
Comments 3:Introduction should briefly state the results of ref 25 and provide a stronger rationale for study in MM.
Response 3: We have revised the Introduction to better highlight the findings of Ref 25 (our prior work on UNC13B in arsenic trioxide resistance) and to provide a clearer rationale for exploring its role in MM.
Comments 4:Methods: more details on lentivirus transduction of MM cells. Negative control? Transient of stable?
Response 4: We thank the reviewer for this helpful comment and apologize for not making these details sufficiently clear in the original submission. In the revised manuscript, we have explicitly stated that ARD multiple myeloma cells were stably transduced with UNC13B-targeting shRNA lentivirus, and that a non-targeting scrambled shRNA construct was used as the negative control. Stable transduction was achieved by puromycin selection (2 μg/mL for 72 h), generating a polyclonal pool of puromycin-resistant cells, which were then used for all subsequent assays.
Comments 5: Fig. 1A: qPCR result compared to what? Non-cancer cells? The y-axis values needs further definition. Why state that U266 and RPMI is overexpression? Fig. 1B: which cell line? There is evidence in databases that UNC13B is often overexpressed in normal, not cancer cells. Please discuss.
Response 5: Thank you for your insightful comments. We have revised the figure legend to more clearly define the comparison group used in the qPCR analysis. Specifically, U266 cells were used as the reference for relative expression calculation in Fig. 1A, and the Y-axis represents the fold-change in UNC13B mRNA levels relative to U266, calculated using the 2^-ΔΔCt method. This has been clarified in the revised figure legend.
We have also updated the Y-axis label in both Fig. 1A and Fig. 1B to read "Relative UNC13B mRNA expression (fold change)" to ensure accurate and standardized representation of the data.
We acknowledge that our statement regarding “overexpression” may have been misleading and have revised it to indicate that ARD cells exhibit relatively higher UNC13B expression compared to the other MM cell lines tested.
Regarding UNC13B expression in normal versus malignant cells, we have now included a discussion in the revised manuscript that acknowledges existing database evidence suggesting that UNC13B may also be expressed in certain normal tissues. We clarify that our study focuses on comparing UNC13B expression across multiple myeloma cell lines, not between malignant and normal plasma cells.
Comments 6:Line 193-194: Only increase in S not G2M.
Response 6: Thank you for pointing this out. Our original flow cytometry results already showed that UNC13B knockdown led to a significant increase in the S phase population, while changes in the G2/M phase were not statistically significant. However, this was inaccurately described in the Results section of the original manuscript. We have corrected the sentence to accurately state that only the S phase population increased significantly, with no significant change in the G2/M phase.
Comments 7: Fig. 3 needs quantification of results.
Response 7: Thank you for your valuable suggestion. We have added quantitative data for the colony formation assay in Fig. 3, including the number of colonies formed in each group (shUNC13B vs. scramble control), along with mean ± SD and statistical significance (p-value). The experimental procedure, cell seeding number, and number of replicates have also been detailed in the revised figure legend.
Comments 8: Fig. 5A: histogram y-axis should be measurement of apoptosis (not mRNA).
Response 8: Thank you for your careful review. We sincerely apologize for the oversight in Figure 5A. You are absolutely correct — the Y-axis was incorrectly labeled as mRNA expression, whereas it should represent the percentage of apoptotic cells as measured by Annexin V-APC staining and flow cytometry. We have corrected the Y-axis label in the revised figure to accurately indicate Apoptotic cells (%) and updated the figure accordingly. Thank you again for bringing this to our attention.
Comments 9: Fig. 5B western blot should include UNC13B. How long were the knockdowns? Analyzed more apoptotic proteins? (anti-apoptotic Bcl-2, Bcl-xL, Mcl-1; pro-apoptotic: Bax., Bak. Cleaved-PARP is a good maker of apoptosis.
Response 9: Thank you for these insightful suggestions. In the revised manuscript, we have updated Fig. 5B to include UNC13B protein levels, thereby confirming the knockdown efficiency at the protein level. The knockdown was maintained for 72 hours post-lentiviral transduction before Western blot analysis.
In addition, we expanded the analysis to include key apoptosis-related proteins. Specifically, we examined cleaved-PARP and Bax expression in ARD cells following UNC13B knockdown. Our results showed that UNC13B protein expression was markedly reduced in the shRNA group compared to the scrambled control. Cleaved-PARP levels were substantially elevated, and Bax expression was increased, consistent with enhanced apoptotic signaling. These findings provide further evidence that UNC13B knockdown promotes apoptosis in multiple myeloma cells.
Comments 10: Define AKR7A3, PINK1 in results, not discussion. Other cell proliferation proteins CDK1, cyclin A, B, E, p21/p27 analyzed?
Response 10: We now provide a brief description of AKR7A3 and PINK1 in the Results section upon their first mention.
We appreciate the suggestion to analyze additional proliferation markers. In the revised manuscript, we added p21 (CDKN1A) to the panel given its direct inhibitory role on the CDK2–Cyclin E/A axis, which aligns mechanistically with our observation of reduced CDK2 and impaired proliferation. Western blot analysis showed an increase of p21 following UNC13B knockdown , supporting a model in which UNC13B silencing suppresses cell-cycle progression via induction of CDK inhibition.
Comments 11: Discussion organization needs major revision. Instead of focusing on western blot targets, should provide more background on UNC13B and why it is considered important in MM. The consequences of UNC13B knockdown should be discussed relative to these targets. The links are not clear.
Response 11: Thank you for this valuable suggestion. We have expanded the Discussion section to include additional background on the role of UNC13B (Munc13-2) in the nervous system and its mechanistic connection to cancer biology. Specifically, UNC13B is a key vesicle-priming factor essential for calcium-dependent exocytosis in neurons and in large dense-core vesicle release from neuroendocrine cells. Beyond neurotransmitter release, Munc13-2–dependent exocytosis has also been implicated in the secretion of bioactive molecules such as cytokines (e.g., IL-6, IL-8), growth factors (e.g., VEGF, TGF-β), proteases (e.g., MMP2, MMP9), and extracellular vesicles carrying integrins and tetraspanins. These secreted factors play key roles in modulating the tumor microenvironment, promoting proliferation, angiogenesis, invasion, and immune evasion. We have added this discussion to highlight that UNC13B may contribute to myeloma progression, at least in part, by regulating vesicle-mediated release of tumor-promoting factors.
Comments 12: More background on UNC13B in neurology and how this relates to cancer. What proteins are secreted that are regulated by UNC?
Response 12: We agree with the reviewer that UNC13B is not uniformly overexpressed across cancers and is indeed highly expressed in normal brain tissue, consistent with its neuronal function. We have revised the Discussion to clarify that the oncogenic relevance of UNC13B in multiple myeloma is likely determined by functional dependency and pathway coupling rather than absolute expression levels. In our model, even in the absence of extreme overexpression, UNC13B knockdown significantly reduced proliferation and increased apoptosis, indicating a critical role in maintaining tumor cell viability. We have also discussed that the expression–function relationship of UNC13B may be context dependent, influenced by its regulation of mitochondrial homeostasis, cell cycle progression, and secretion of tumor-promoting factors, and may thus differ between normal tissue and malignant cells.
Comments 13: There does not appear to be a correlation with UNC13B overexpression in cancer; often more highly expressed in normal tissue.
Response 13: We agree with the reviewer that UNC13B is not uniformly overexpressed across cancers and is indeed highly expressed in normal brain tissue, consistent with its neuronal function. We have revised the Discussion to clarify that the oncogenic relevance of UNC13B in multiple myeloma is likely determined by functional dependency and pathway coupling rather than absolute expression levels. In our model, even in the absence of extreme overexpression, UNC13B knockdown significantly reduced proliferation and increased apoptosis, indicating a critical role in maintaining tumor cell viability. We have also discussed that the expression–function relationship of UNC13B may be context dependent, influenced by its regulation of mitochondrial homeostasis, cell cycle progression, and secretion of tumor-promoting factors, and may thus differ between normal tissue and malignant cells.
Comments 14: Are there any drugs available that can be used to test? Otherwise, the potential clinical relevance of this study is questionable.
Response 14: Thank you for raising the important point regarding the lack of drugs targeting UNC13B. We acknowledge that, at present, there are no clinically available agents that specifically inhibit UNC13B. To address this limitation, we have now added a section in the Discussion highlighting the pharmacological landscape:
A preclinical study identified a class of compounds, termed bexins, that inhibit Munc13-4-mediated granule fusion in immune cells, demonstrating that pharmacological inhibition of Munc13 functions is indeed feasible.
Additionally, bryostatin 1, a known C1-domain ligand, has been shown to interact with Munc13-1, suggesting that targeting the C1/C2 domains may represent a viable strategy to modulate UNC13 activity pharmacologically.
While these examples do not directly target UNC13B, they illustrate potential indirect or domain-related approaches for disrupting UNC13B-mediated exocytosis.
Comments on the Quality of English Language: English writing is acceptable, with only minor improvemnts required.
Response : Thank you for the positive assessment. We have carefully revised the manuscript to address minor language issues and improve clarity and flow. In this revision, we corrected typographical and punctuation errors, tightened several sentences for concision, and ensured consistent use of terminology and abbreviations throughout.
Reviewer 2 Report
Comments and Suggestions for Authors
The manuscript entitled “Inhibiting UNC13B suppresses cell proliferation by upregulating 2 the apoptotic pathway in multiple myeloma” highlighted that UNC13B contributes to MM cell survival and 34 proliferation, potentially through activation of oncogenic signaling pathways. The manuscript is quite interesting with clear methodologies and results, but it lacks a comprehensive discussion, which needs to be completely revised.
Comments on the Quality of English LanguageThe manuscript entitled “Inhibiting UNC13B suppresses cell proliferation by upregulating 2 the apoptotic pathway in multiple myeloma” highlighted that UNC13B contributes to MM cell survival and 34 proliferation, potentially through activation of oncogenic signaling pathways. The manuscript is quite interesting with clear methodologies and results, but it lacks a comprehensive discussion, which needs to be completely revised.
Author Response
Comments and Suggestions for Authors: The manuscript entitled “Inhibiting UNC13B suppresses cell proliferation by upregulating 2 the apoptotic pathway in multiple myeloma” highlighted that UNC13B contributes to MM cell survival and 34 proliferation, potentially through activation of oncogenic signaling pathways. The manuscript is quite interesting with clear methodologies and results, but it lacks a comprehensive discussion, which needs to be completely revised.
Response : We sincerely thank the reviewer for this constructive comment regarding the need for a more comprehensive discussion. In the revised manuscript, we have substantially expanded and reorganized the Discussion section to provide a broader and deeper interpretation of our findings, as follows:
Neurological background and cancer relevance: We now summarize the established role of UNC13B in synaptic vesicle priming and neurotransmitter release, and discuss how its vesicle fusion machinery may be co-opted by malignant cells to regulate the secretion of tumor-promoting cytokines and growth factors. This addition also addresses the potential molecular parallels between neuronal exocytosis and cancer cell secretory pathways.
Secreted proteins regulated by UNC13B: We have incorporated evidence that UNC13B-related exocytosis in non-neuronal cells can modulate the release of interleukins, chemokines, and extracellular vesicles, which are relevant to the multiple myeloma tumor microenvironment.
Expression patterns in cancer versus normal tissues: We acknowledge that UNC13B can be highly expressed in some normal tissues and have discussed the implications of this for therapeutic targeting, as well as the need for tumor-selective inhibition strategies.
Clinical relevance and druggability: We added a section discussing the current lack of direct pharmacological inhibitors for UNC13B, and reviewed preclinical examples of small molecules that target related Munc13 family members or functional domains, illustrating that therapeutic modulation is feasible and represents a promising direction for future research.
Integration with our experimental data: We relate these broader points back to our current findings, highlighting that UNC13B knockdown not only suppresses proliferation and induces apoptosis in MM cells, but also disrupts signaling pathways (PINK1, AKR7A3, CDK2, p21) that are central to both survival and cell-cycle progression.
These additions collectively strengthen the manuscript by situating our results within a broader biological and translational context, as requested.
Reviewer 3 Report
Comments and Suggestions for Authors
Thank you for allowing me the opportunity to review your manuscript. The authors have chosen to conduct a study on Inhibiting UNC13B suppress cell proliferation by upregulating the apoptotic pathway in multiple myeloma a unique contribution to the existing literature.
However, some revisions are needed before publication.
The paper should be revised in terms of language. Word errors, grammatical spacing, and punctuation marks should be taken into account. . Furthermore, the guidelines recommend that abbreviations should be defined at the first occurrence in the text and used consistently thereafter. Please make sure to follow this rule throughout the paper.
The purpose of this study should be clearly stated, which gap in the literature it will fill.
The method by which gene suppression efficiency was assessed 24 hours after transfection should be specified. Why was 24 hours chosen?
Quantitative PCR (qPCR) was used to assess UNC13B mRNA expression in U266, 104 ARD, RPMI 8226, and other multiple myeloma cell lines. Other cell lines should be specified.
Detailed information should be provided about the kits and protocols used for all experiment.
All analyses performed should be specified in which cell line they were performed.
Were only ARD cells transfected? Why were only these cells selected for further analysis? This should be stated.
PINK1, CDK2, AKR7A3, and Bim, these molecules were chosen based on what methods?
Bioinformatics analyses included in the abstract should be written in the material and methods section, and detailed results should be given in the discussion and material and methods section.
The discussion section is not well presented. Based on the results obtained, the discussion should be rewritten by reconsidering the data in the literature.
These suggestions will prove helpful in enhancing the quality of your manuscript.
Author Response
Comments 1: The paper should be revised in terms of language. Word errors, grammatical spacing, and punctuation marks should be taken into account. . Furthermore, the guidelines recommend that abbreviations should be defined at the first occurrence in the text and used consistently thereafter. Please make sure to follow this rule throughout the paper.
Response 1: We thank the reviewer for this valuable comment. The manuscript has been carefully revised to correct word usage errors, grammatical issues, spacing, and punctuation throughout. We have also ensured that all abbreviations are defined at their first occurrence in the text and are used consistently thereafter, in accordance with the journal’ s guidelines. These changes aim to improve the overall readability and compliance with formatting requirements.
Comments 2: The purpose of this study should be clearly stated, which gap in the literature it will fill.
Response 2: We thank the reviewer for this valuable suggestion. In the revised manuscript, we have explicitly stated the purpose of the study and the gap in the literature it aims to address in the Introduction section. Specifically, at the end of the Introduction, we now clearly indicate that the present study aims to determine whether UNC13B plays a functional role in multiple myeloma, to elucidate its effects on cell proliferation and apoptosis, and to explore the underlying molecular mechanisms. This addition highlights the novelty of our work in filling the knowledge gap regarding the role of exocytosis-related proteins in MM pathogenesis and their potential as therapeutic targets.
Comments 3: The method by which gene suppression efficiency was assessed 24 hours after transfection should be specified. Why was 24 hours chosen?
Response 3: We appreciate the reviewer’s attention to this detail. In the original manuscript, the time point for assessing knockdown efficiency was mistakenly stated as 24 hours post-selection due to an oversight. We have now corrected this to 72 hours post-selection, which is the standard time point at which shRNA-mediated suppression reaches near-maximal efficiency before performing protein expression and functional assays. This change has been updated in the revised Methods section accordingly.
Comments 4: Quantitative PCR (qPCR) was used to assess UNC13B mRNA expression in U266, 104 ARD, RPMI 8226, and other multiple myeloma cell lines. Other cell lines should be specified.
Response 4: We thank the reviewer for pointing out this ambiguity. In the original manuscript, the phrase “and other multiple myeloma cell lines” was mistakenly included. In fact, only three multiple myeloma cell lines were analyzed in this study: U266, ARD, and RPMI 8226. The Methods section has been revised accordingly to accurately reflect the experimental design.
Comments 5: Detailed information should be provided about the kits and protocols used for all experiment.
Response 5: We appreciate the reviewer’s valuable suggestion. In the revised manuscript, we have provided detailed information about all experimental kits, reagents, and instruments used, including manufacturers and catalog numbers, for each method described. This includes cell culture media and supplements, transfection reagents, RNA extraction and cDNA synthesis kits, qPCR reagents, cell proliferation and apoptosis assay kits, flow cytometry staining reagents, and Western blotting materials. These details have been added to the Materials and Methods section to ensure full reproducibility of our experiments.
Comments 6: All analyses performed should be specified in which cell line they were performed.
Response 6: All functional experiments (shRNA knockdown, CCK-8, colony formation, apoptosis, and Western blot) were conducted in ARD cells. This is now clearly stated throughout the manuscript.
Comments 7: Were only ARD cells transfected? Why were only these cells selected for further analysis? This should be stated.
Response 7: We thank the reviewer for this valuable comment. In our initial screening (Fig. 1A), we found that ARD cells exhibiting the highest UNC13B mRNA expression level (3.07 ± 0.44). To maximize the likelihood of observing measurable phenotypic effects following gene silencing, we selected ARD cells for subsequent transfection and functional assays. We have revised the Results section accordingly to clarify this rationale.
Comments 8: PINK1, CDK2, AKR7A3, and Bim, these molecules were chosen based on what methods?
Response 8: We appreciate the reviewer’s question regarding the rationale for selecting PINK1, CDK2, AKR7A3, and Bim for analysis. These proteins were chosen based on our preliminary transcriptomic data and published literature, which suggested their potential association with UNC13B function:
PINK1 is a mitochondrial serine/threonine kinase involved in quality control and stress signaling, relevant to apoptosis regulation and cell survival.
CDK2 is a key regulator of the G1/S transition and cell cycle progression, processes that were affected by UNC13B knockdown in our study.
AKR7A3 participates in aldehyde/ketone detoxification and has been implicated in cancer-related metabolic and signaling pathways.
Bim is a pro-apoptotic BCL-2 family member that mediates intrinsic apoptosis.
In the revised manuscript, we expanded our protein panel to include cleaved-PARP, Bax, p21, and PKC, providing a more comprehensive assessment of mitochondrial homeostasis, cell cycle regulation, and apoptotic signaling after UNC13B silencing.
Comments 9: Bioinformatics analyses included in the abstract should be written in the material and methods section, and detailed results should be given in the discussion and material and methods section.
Response 9: We appreciate the reviewer’s comment. In the original submission, we mistakenly included brief descriptions of bioinformatics analyses in the abstract. As these analyses were not part of the final study, we have removed all related content from the abstract, Materials and Methods, and Discussion sections to maintain consistency and accuracy
Comments 10: The discussion section is not well presented. Based on the results obtained, the discussion should be rewritten by reconsidering the data in the literature.
Response 10: We appreciate the reviewer’s suggestion regarding the Discussion section. Following this comment, we have substantially revised the Discussion by integrating our experimental findings with relevant literature to provide a clearer interpretation of the results. The updated section now includes a more comprehensive comparison with previous studies, highlights the potential mechanisms underlying our observations, and discusses the translational implications of UNC13B in multiple myeloma.
Round 2
Reviewer 1 Report
Comments and Suggestions for Authors
Responses are appropriate, except for lines 252-253 where "increased number of cells in S phase, not in G2M".
Author Response
Comments 1: Responses are appropriate, except for lines 252-253 where "increased number of cells in S phase, not in G2M".
Response: We thank the reviewer for pointing this out. In the first revision we corrected the main text but inadvertently left an outdated sentence in the Fig. 4 caption. We have now revised the caption to state that G1 is significantly reduced and S phase is significantly increased, with no statistically significant change in G2/M.
Revised caption: “Fig. 4. Effect of UNC13B gene ablation on the cell cycle in ARD cells. Five days after the transfection with shRNA lentiviral vector, the shUNC13B group had a reduced number of ARD cells in the G1 phase (P < 0.0001) and an increased number of cells in the S phase (P < 0.0001), with no statistically significant change observed in the G2/M phase (ns).”
All changes are shown using Microsoft Word Track Changes in the revised manuscript (lines 252–253).
Reviewer 2 Report
Comments and Suggestions for Authors
The authors have revised the manuscript and answered the queries raised by the reviewer. It may be accepted for publication
Author Response
Comments 1: The authors have revised the manuscript and answered the queries raised by the reviewer. It may be accepted for publication
Response: We sincerely thank the reviewer for the positive evaluation and recommendation for acceptance. We have rechecked the revised manuscript to ensure that all corrections are accurately reflected across the text, figures, and legends and that the formatting adheres to the journal’s guidelines. Although no further changes were requested, we performed minor language polishing for clarity without altering the results or conclusions. We greatly appreciate the reviewer’s time and consideration.
Reviewer 3 Report
Comments and Suggestions for Authors
The paper should be revised in terms of language. Word errors, grammatical spacing, and punctuation marks should be taken into account.Furthermore, the guidelines recommend that abbreviations should be defined at the first occurrence in the text and used consistently thereafter. Please make sure to follow this rule throughout the paper.Figures should be understood more clearly.
I hope the suggestions for the better of this prepared manuscript will be useful to you.
Comments on the Quality of English LanguageThe paper should be revised in terms of language. Word errors, grammatical spacing, and punctuation marks should be taken into account.
Author Response
Comments : The paper should be revised in terms of language. Word errors, grammatical spacing, and punctuation marks should be taken into account.Furthermore, the guidelines recommend that abbreviations should be defined at the first occurrence in the text and used consistently thereafter. Please make sure to follow this rule throughout the paper.Figures should be understood more clearly.
I hope the suggestions for the better of this prepared manuscript will be useful to you.
Comments on the Quality of English Language:
The paper should be revised in terms of language. Word errors, grammatical spacing, and punctuation marks should be taken into account.
Response:
We sincerely thank the reviewer for these helpful comments. We conducted a line-by-line revision to improve English usage, formatting, and figure readability, and we implemented the following changes throughout the manuscript. All edits are visible in the Tracked Changes version, with a clean version provided as well.
Language polishing & consistency
Corrected grammar, diction, and punctuation (e.g., added/removed commas appropriately; fixed spacing; standardized American English usage).
Standardized scientific phrasing and hyphenation (e.g., cell-cycle, S-phase, C2-domain-dependent, anti-proliferative).
Improved flow/conciseness by removing duplicated or redundant sentences where appropriate.
Abbreviations: first mention & global consistency
Ensured every abbreviation is defined at first occurrence and used consistently thereafter (e.g., quantitative real-time PCR (qPCR); 2^−ΔΔCt method; OD450; ns = not significant; etc.).
Nomenclature for genes/proteins
Followed HGNC-style conventions: gene symbols italicized when referring to gene/mRNA or gene-level manipulations (e.g., UNC13B knockdown, UNC13B mRNA), and proteins in roman (UNC13B protein, Munc13-2, Bim). This is now consistent across text and figure legends.
Units, statistics, and style
Standardized units/symbols and spacing: μL, μg/mL, °C, % COâ‚‚, etc..
Regularized ranges and citation spans with en-dashes (e.g., 7–8 years; [21–23]).
Figure clarity and legends
Added replicate information and statistical annotations to legends across figures.
Specific updates:
Fig. 4: Legend corrected to “decreased G1, increased S-phase, and no significant change in G2/M,” aligning with the main text.
Legends now explicitly define ns and the statistical test used.
Section-specific refinements:
Abstract/Results/Discussion: Removed duplicated phrasing, corrected minor diction (e.g., “several human multiple myeloma cell lines” instead of “multiple human multiple myeloma”), and smoothed transitions without altering scientific content.
Methods (qPCR): Standardized to “quantitative real-time PCR (qPCR)” and the 2^−ΔΔCt calculation.
Discussion: Corrected punctuation and hyphenation, and improved readability while preserving references and conclusions.
We greatly appreciate the reviewer’s guidance. These comprehensive edits have improved the clarity, readability, and consistency of the manuscript, including clearer figure presentation and fully consistent usage of abbreviations and nomenclature.